# Impacts of Strong ENSO Events on Fish Communities in an Overexploited Ecosystem in the South China Sea

**DOI:** 10.3390/biology12070946

**Published:** 2023-07-01

**Authors:** Miao Li, Youwei Xu, Mingshuai Sun, Jiajun Li, Xingxing Zhou, Zuozhi Chen, Kui Zhang

**Affiliations:** 1South China Sea Fisheries Research Institute, Chinese Academy of Fishery Sciences, Guangzhou 510300, China; 2Key Laboratory for Sustainable Utilization of Open-Sea Fishery, Ministry of Agriculture and Rural Affairs, Guangzhou 510300, China; 3College of Marine Sciences, Shanghai Ocean University, Shanghai 201306, China

**Keywords:** fish community, biodiversity, Beibu Gulf, GAM, *Trachurus japonicus*, *Decapterus maruadsi*

## Abstract

**Simple Summary:**

Impacts of climate change on fisheries resources and biodiversity are being increasingly reported. An increased frequency and intensity of extreme ocean climate events can be expected to affect Beibu Gulf marine ecosystems and fishery productivity. We examine Beibu Gulf fisheries resource survey data before and after ENSO events, and compare changes in fish biodiversity, dominant species, and community distributions between El Niño and La Niña events. Relationships between the abundance of dominant species and environmental variables are discussed, and how environmental changes affect fish populations are evaluated.

**Abstract:**

To better understand how fish communities respond to environmental changes under extreme climate events, we examine changes in fish communities in Beibu Gulf during strong El Niño and La Niña events. Strong La Niña and El Niño events affect the composition, abundance, and distribution of fish communities in Beibu Gulf. Fish community distribution and composition change before and after La Niña and El Niño events, and dominant species within them change with stable fishing intensity. The abundance and distribution of small pelagic fish such as Japanese jack mackerel (*Trachurus japonicus*) and Japanese scad (*Decapterus maruadsi*) are the most affected. Using a generalized additive model (GAM), we explore relationships between the abundance of *T. japonicus* and *D. maruadsi* and a suite of environmental variables. The GAM results revealed that sea surface salinity and sea surface temperature best explain changes in catch per unit effort of these two species during a La Niña event; depth, sea surface temperature, and mixed layer depth during an El Niño event. The results obtained in this study will offer support for implementing more-accurate, scientific fisheries management measures.

## 1. Introduction

The El Niño–Southern Oscillation (ENSO) is a major sea surface temperature (SST) anomaly triggered by ocean–atmospheric interactions, represented by ocean warming (El Niño) and cooling (La Niña) events [1]. ENSO events are measured by the Oceanic Niño Index (ONI), which indicates the intensity of El Niño or La Niña events [2]. Climate change exacerbates the occurrence of ENSO and severe weather events, which in turn affect marine fish communities [3].

ENSO events affect marine climate, hydrology, and the ecological status of species at various spatial and temporal scales [4,5]. However, topographic constraints on species dispersal, biological interactions, and species characteristics can contribute to lags between changes in climate and fish communities [6,7,8,9,10]. Changes in environmental conditions can affect species growth, mortality, migration, spawning [11,12,13], and fish community composition and distribution [14,15,16]. Changes in the spatio-temporal distributions of species can also affect fisheries catch and economies [17].

Beibu Gulf is a traditional, historically biodiversity-rich marine fishing ground in the northwestern South China Sea [18]. In recent decades, disturbances such as climate change and overfishing have impacted the productivity and biodiversity of marine ecosystems in this region [19,20]. To protect marine fisheries, management policies such as “double control” of fishing vessels (limiting their number and power), a total management system (limiting fishing quota allocation and protecting juvenile fish), and seasonal fishing moratoriums have been implemented [21]. These policies have led to a gradual balance among fishing capacity, catch amount, fisheries resource carrying capacity, and fishing intensity in these waters has stabilized or even decreased [22,23]. Despite this, ongoing anthropogenic disturbance and increasingly intense environmental changes mean that the recovery of fishery resources in the South China Sea will take time. Not only has overexploitation of fish resources in Beibu Gulf changed fish community composition [18,21,24,25], but ecosystems within it that are already compromised by fisheries are further affected by climate change [18,26], especially given the increased frequency and intensity of climate events [5]. Extreme events associated with climate change increase uncertainty for fisheries management. Therefore, it is important to understand how fishery resources respond to climate change in marine ecosystems, especially those that have been overexploited, such as the Beibu Gulf.

ENSO events affect sea levels and surface temperatures, primary productivity, and ocean circulation in the South China Sea [27,28,29]. Cascading changes in habitat may affect the growth, mortality, fecundity, spawning distribution, and movement of fish in Beibu Gulf. Small pelagic fishes, which occupy an important ecological niche in Beibu Gulf, are particularly highly sensitive to climate change [12,16]. Relationships between ENSO and fish communities in Beibu Gulf are not well understood. We examine historical fisheries data for this region and identify changes in fish species composition, diversity, abundance, and communities before and after El Niño and La Niña events. To investigate which changes in fish community structure are caused mainly by climate change (as opposed to fisheries impacts), we examined data from a summer fishing closure period. A generalized additive model (GAM) was used to identify relationships between fishery resources and environmental variables [30]. We evaluate how the environment affects pelagic fish resources in Beibu Gulf, and what environmental variables most strongly affect the abundance of these species. Our results provide a scientific basis for improved management of fishery resources in this dynamic environment.

## 2. Materials and Methods

### 2.1. Data Collection

#### 2.1.1. Oceanic Niño Index (ONI)

Time series ONI data were obtained from the National Oceanic and Atmospheric Administration website (https://origin.cpc.ncep.noaa.gov/products/analysis_monitoring/ensostuff/ONI_v5.php, accessed on 5 July 2022). ONI indicate that a strong La Niña event occurred from 2007 to 2008 (Figure 1a) and a strong El Niño event occurred from 2015 to 2016 (Figure 1b).

#### 2.1.2. Survey Data

Fish trawl survey data were obtained from the South China Sea Fisheries Research Institute, Chinese Academy of Fishery Sciences, for the month of July (summer), for the years 2006, 2008, 2014, and 2016. July is in the summer fishing moratorium in the South China Sea, during which time fishing effort in Beibu Gulf can be considered well controlled and negligible. Data for selecting the fishing moratorium period are intended to reduce the impact of fishing, mainly considering the impact of climate change. Additionally, small pelagic fishes, which occupy an important ecological niche in Beibu Gulf, are particularly highly sensitive to climate change [12,16]. The breeding period for small pelagic species in the Beibu Gulf, represented by T. japonicus and D. maruadsi, is mainly in winter. ENSO events affect their recruitment processes primarily by altering environmental factors. The fish distribution and density data from July provide a visual indication of the impact of their winter recruitment. Bottom trawl surveys were performed using a 441 kW survey vessel with a net of 76 m × 53.79 m (headline 34 m), with 20 cm mesh, and 3.9 cm cod-end mesh. Each year, the same 52 stations (Figure 2) were sampled by trawls towed for 1 h at 3.5 kn. Upon retrieval on deck, catch was sorted, identified by species, and counted; the total weight of each fish was determined.

### 2.2. Environmental Data

Online data sources, units, and the spatial resolutions of remote-sensed data are detailed in Table 1. Sea surface salinity (SSS), mixed layer depth (MLD), sea surface chlorophyll a (Chl-a), and dissolved oxygen (DO) data were obtained from the Copernicus Marine Environmental Monitoring Service (CMEMS). SST data were obtained from the Moderate Resolution Imaging Spectroradiometer (MODIS). Satellite precipitation data (Pre) were acquired from the Tropical Precipitation Measurement Mission (TRMM). Bathymetric data were obtained from trawl surveys.

### 2.3. Data Analyses

#### 2.3.1. Dominant Fish Species

We used the index of relative importance (IRI) and its percentage (pIRI) to analyze the status of abundance and weight of fish species in Beibu Gulf fish communities [31]. These two indexes are calculated as follows:(1)IRI=(N+W)∗F
(2)pIRI=IRI/∑IRI
where *N* is the percentage of the tail number of a species to the total tail number; *W* is the percentage of biomass to the total weight of a species; and *F* is the frequency of occurrence of the fish species during the survey. The pIRI is the percentage of a single species to total IRI. Species were divided by dominance: IRI ≥ 1000 (dominant), 500 ≤ IRI < 1000 (important), 100 ≤ IRI < 500 (common), 10 ≤ IRI < 100 (uncommon), and IRI < 10 (rare) [32].

#### 2.3.2. Diversity

We analyzed pre- and post-ENSO event fish diversities using Margalef’s richness (*D*′), Shannon–Weiner diversity (*H*′), and Pielou’s evenness (*J*′) indexes, calculated following Margalef et al. [33], Pielou et al. [34], and Wilhm et al. [35]:(3)$$D^{\prime }=\left(S-1\right)/lnR$$
(4)$$H^{\prime }=-\sum_{i=1}^{S}P_{i}lnP_{i}$$
(5)$$J^{\prime }=H^{\prime }/lnS$$
where *S* is the total count of fish species in a sample; *R* is the total count of fish individuals; and *P_i_* is the biomass proportion of the *i*-th fish. Analysis of variance (ANOVA) was used to compare differences in fish diversity in summer surveys before and after ENSO events.

#### 2.3.3. Fish Community Composition

To reduce deviations in community composition caused by rare species, we removed those with a catch ratio <0.1% and collected from <2% of stations from analysis. Data were fourth-root transformed to construct a Bray–Curtis similarity coefficient matrix. Fish community structure was examined using cluster analysis and non-metric multidimensional scale ranking (NMDS). Based on similarity analysis (ANOSIM), differences in fish species composition before and after ENSO events were assessed. Data analyses were performed using the ‘vegan’ package in R version 4.3.0. The spatial distribution of abundance for certain commercially exploited species was drawn using ArcGIS10.8 software.

#### 2.3.4. GAM Model Development

Relationships between the spatial and temporal distribution of fishery resources and environmental variables is complex, non-linear, and non-additive [36]. Generalized additive models can better identify relationships between response variables and multiple explanatory variables, have a high degree of flexibility in model construction, and are widely used in the qualitative analysis of relationships between fisheries resources and the marine environment [37,38].

The GAM formula is:(6)$$Y=\alpha +\sum_{j=1}^{n}f_{i}(x_{i})+\varepsilon$$
where *Y* is the response variable to be modeled (CPUE); *α* is the intercept; *f_i_*(*x_i_*) is the smoothing function of the covariate, and *ε* is an error term [39]. We used GAM to explore which environmental variables most affect fish abundance. Transformed (log (Abundance + 1), so that residuals approached a normal distribution) resource abundance data for certain commercially exploited fish species were used as a response variable to simulate the response of fishes to seven environmental predictors (SST, SSS, DO, MLD, Chl-a, Pre, Depth). A variance inflation factor (VIF) was calculated to assess multiple collinearities, and to select environmental variables for inclusion in the analysis. Variables with VIF < 10 were selected as model predictors. Log-transformed abundance data were analyzed with stepwise regression, and the fit was judged using the value of the Akaike Information Criterion (AIC); the lower the value, the better the model fit [40]. The generalized cross-validation (GCV) of the model was used to obtain an optimal model and its response variables. Analysis was performed using the ‘vegan’ package in R version 4.3.0.

## 3. Results

### 3.1. Fish Species Composition

Prior to a July 2006 La Niña event, 184 fish species were identified. Following a July 2008 La Niña event, 212 fish species were identified. Of these species, 145 occurred in both surveys and accounted for 97.53% of the total survey biomass. Dominant species (IRI > 1000) in July 2006 and 2008 were glowbelly (*Acropoma japonicum*) and *T. japonicus*, and *D. maruadsi* was also dominant in 2008 (Table 2). After the La Niña event, the catch biomass and abundance of dominant and important species changed significantly, with abundance of warm-water fish, represented by *T. japonicus* and *D. maruadsi*, increasing, and that of warm-temperate fish, represented by *A. japonicum*, decreasing.

During July 2014 and 2016 (pre- and post an El Niño event), 194 and 191 fish species were identified, respectively. Of these, the 131 species found in both surveys accounted for 97.89% of the total biomass. While dominant species in July 2014 comprised *A. japonicum*, *T. japonicus*, and threadfin porgy (*Evynnis cardinalis*), those in 2016 comprised *T. japonicus*, *A. japonicum*, Japanese butterfish (*Psenopsis anomala*), *E. cardinalis*, and *D. maruadsi*. Following the El Niño event, the proportion of dominant species increased, the relative abundance of *A. japonicum* and *E. cardinalis* decreased, *T. japonicus* became the dominant taxon, and replacement of dominant species was evident (Table 2).

### 3.2. Fish Diversity and Community Structure

Trends in fish community diversity before and after the ENSO event are illustrated in Figure 3. Of indices, *D*′ reduced significantly following a La Niña event (*p* < 0.05), primarily in the northeastern gulf, but there was no significant change in pre- and post-El Niño values (*p* > 0.05). Pre- and post-La Niña *H*′ did not differ significantly throughout Beibu Gulf (*p* > 0.05), but the decrease in northeastern coastal gulf waters was significant (*p* < 0.05). There was a significant difference in *H*′ after and before the occurrence of El Niño (*p* < 0.05). Pre- and post-La Niña *J*′ did not differ significantly throughout Beibu Gulf (*p* > 0.05), but there was a significant decrease in the northeastern coastal region (*p* < 0.05). Differences between *J*′ after and before an El Niño event were significant (*p* < 0.05), with post-El Niño event values tending to decrease.

### 3.3. Distributions of Fish Assemblages

Cluster analysis and NMDS revealed spatial and temporal changes in fish assemblages (Figure 4). Because NMDS stress values for each year are <0.2, we deem results to be of explanatory significance.

Two main fish communities occurred pre- and post-La Niña events (Figure 5). Community I occurred primarily in coastal northeastern Beibu Gulf, and mostly comprised small fishes such as *Leiognathus*, banded scad (*Alepes djedaba*), *T. japonicus*, and *E. cardinalis*. The second, widely distributed community (II) occurred primarily in southern–central gulf waters, and was characterized by *A. japonicum*, *T. japonicus*, *E. cardinalis*, and big-head pennah croaker (*Pennahia macrocephalus*). Abundance and distribution of *T. japonicus* and *D. maruadsi* increased after a La Niña event, but *A. japonicum* decreased, and other species were disadvantaged by competition. ANOSIM revealed that pre- and post-La Niña fish communities in Beibu Gulf differ significantly (*p* < 0.05).

Three main fish communities occurred pre- and post-El Niño events (Figure 5). Community I occurred primarily in coastal eastern gulf waters, and was dominated by *E. cardinalis*, *P. macrocephalus*, and rabbitfish (*Siganus oramin*). Community II occurred primarily in the southwestern gulf (below Bailongwei Island), and was characterized by species of *Leiognathus*. Community III occurred primarily in southern–central gulf waters, and was characterized by *T. japonicus*, *A. japonicum*, *P. anomala*, *E. cardinalis*, and *D. maruadsi*. Fish abundance and biomass in northeastern gulf coastal waters decreased significantly following an El Niño event. The abundance of *T. japonicus* increased and that of *A. japonicum* decreased after El Niño, and the distribution of dominant species was similar.

### 3.4. GAM Analysis

Significant differences in total biomass and abundance were apparent pre- and post-La Niña events, primarily for small pelagic fish such as *T. japonicus* and *D. maruadsi*. After removing these two species, there was no significant difference in catch between years (*p* > 0.05). While fisheries resources decreased following an El Niño event, the difference was not significant (*p* > 0.05), and abundances of *T. japonicus* and *D. maruadsi* increased significantly (*p* < 0.05). Because the distributions of fishes depend on environmental conditions within a season, the quantity of fish depends on environmental conditions experienced in the previous season. Winter is important for spawning and nursing of commercially exploited fish such as *T. japonicus* and *D. maruadsi*, and fluctuations in their abundances are closely related to those in their environment.

We used GAMs to assess the abundance of *T. japonicus* and *D. maruadsi* and environmental variables (winter). The Beibu Gulf is located in tropical to subtropical waters with high water temperatures. El Niño events may produce abnormal environmental changes such as elevated water temperature, which can limit the growth, survival, and distribution of fish. La Niña phenomena bring conditions such as low temperature and high primary productivity, which also affect fish populations. To explore the impacts of environmental change under ENSO events, we constructed GAMs to analyze El Niño and La Niña events separately.

Based on minimum corrected AIC and GCV values, the most appropriate model variables (SST, SSS, DO, MLD, Chl-a, Pre, and Depth) were selected using reverse stepwise regression (Table 3).

#### 3.4.1. La Niña

Of the seven environmental variables for which we have data, GAM revealed that SST, Chl-a, Depth, DO, and SSS best explain variation in *T. japonicus* and *D. maruadsi* abundance. The interpretive bias of these variables was 63.2%, with an *R*^2^ of 0.553 (Table 3). Of them, SSS was the most influential, with a relative contribution of 39.4% (in a single model), followed by SST (23%), Depth (13%), DO (10%), and Chl-a (6%). *F*-tests reveal that SST, Depth, DO, and SSS correlate significantly with abundance (*Pr*(*F*) < 0.05). When Chl-a was added to the model, the AIC value further decreased and the cumulative interpretive bias increased, indicating that the model fit increased. Therefore, we retained Chl-a in the model.

The relationship between environmental variables and resource abundance was nonlinear (Figure 6). The effect of SST was positive from 18 to 19.2 °C and 20.4 to 22 °C (Figure 7). While Chl-a from 0.35 to 0.8 mg m^−3^ negatively affected abundance, the effect was positive from 0.8 to 1.2 mg m^−3^. Abundances were mainly concentrated between SSSs of 33.2 and 33.8 (Figure 8), and the effect was positive between 31.8 and 32.1. Depth and DO were positively correlated with abundance; the effect was positive from 20 to 80 m depth and 215–240 mg m^−3^, respectively.

#### 3.4.2. El Niño

GAM revealed that SSS, Pre, MLD, Depth, and SST best explained variation in *T. japonicus* and *D. maruadsi* abundance. The interpretive bias of these variables was 62.7%, with an *R^2^* of 0.571 (Table 3). Of them, Depth had the greatest effect (48.2% relative contribution) on abundance, followed by SST (41.3%), MLD (36.4%), SSS (23.7%), and Pre (12.8%). *F*-tests revealed that SST, Depth, and MLD correlated significantly with abundance (*Pr*(*F*) < 0.05). When Pre and SSS were added to the model, the AIC value decreased, so we retained both variables.

The relationship between environmental variables and resource abundance is shown in Figure 9. SSS positively affected resource abundance from salinities of 31.0–31.7. Within the range of 0.2–0.4 mm·h^−1^, Pre increased gradually, and abundance increased. Abundance was positively affected by Depth from 20 to 75 m and SST from 18 to 25.5 °C (Figure 10). Both *T. japonicus* and *D. maruadsi* occurred mostly at 15–45 m MLD (Figure 11).

## 4. Discussion

Climate variability in the Pacific is largely governed by ENSO [41]. The diversity of ENSO types affects fish populations and marine ecosystems in different ways [5]. While historical survey data indicate that a rich variety of fish species occurred in Beibu Gulf [18,26], fish species richness in this region has trended downward over time [24,42,43]. Natural succession in community structure can cause fluctuations in numbers of fish species, but these fluctuations are not significant [18,32].

The number of fish species in Beibu Gulf in July 2006 was slightly lower than in July 2008, possibly because of the coincidence of the fishery resource survey and a La Niña event provided good habitat, spawning, and feeding conditions for fish species [44], and/or because implementation of fishery resource conservation policies contributed to recovery of fishery resources. The number of fish species in July 2014 and 2016 fluctuated little. We report that the number of dominant fish species increase significantly after a La Niña event, and that *T. japonicus* and *D. maruadsi* occupy a dominant position in the community. With an increase in dominant species, other fish species may be competitively disadvantaged, and species range varies widely. The distributions of fish communities changed, as did species diversity (Figure 3).

Following a La Niña event, *D*′ differed significantly and trended downward. In northeastern gulf waters, changes in species and their distributions caused significant reductions in *H*′ and *J*′ [45]. High Chl-a in northeastern coastal gulf waters increased primary productivity and provided abundant food for juvenile fish; fish community distributions were relatively stable [26]. A La Niña event caused a persistent decrease in gulf water SST (Figure 7). Increasing wind strength also creates vortices and upwellings, drawing nutrient-rich deeper waters to the surface, increasing productivity, and promoting growth and reproduction of fish prey and fishes [46].

An increase in the number of dominant fish species occurred following an El Niño event. The total number of fish reported in each survey varied little, and there was no significant difference in *D*′. Both *H*′ and *J*′ generally trended downward, decreasing significantly in coastal waters, and the abundance of coastal fish in northeastern waters decreased. Frequent tropical cyclones in the South China Sea, wind-induced mixing and upwelling promote primary productivity. Fish production in the northern South China Sea correlates positively with the winter monsoon [47]. We report that a weakening of the winter monsoon during an El Niño event coincided with a decrease in primary productivity, and with fisheries resources trending downward, consistent with Qiu et al. [47].

ENSO-driven changes in the environment affect pelagic fish populations [5]. Changes in the physiological function of fish, and their migration and spawning frequency, affect population structure and distribution [48,49,50]. Changes in the distribution and abundance of many commercial fish species are consistent with the occurrence of an ENSO event [51,52,53]. Small pelagic fishes in Beibu Gulf also showed changes in size and distribution in response to environmental changes caused by an ENSO event.

Small pelagic fish have short life cycles, greater mobility than demersal fish, and are sensitive to changes in their environment. [5] Because the environment affects plankton dynamics, changes can cascade to affect small pelagic fish populations, mainly in their biomass, distribution, spawning, reproduction, and species composition [54,55]. Fishes can adapt to environmental change by migrating to more favorable areas. Changes in SST, SSS, Chl-a, Depth, and DO all affected the abundance of *T. japonicus* and *D. maruadsi* before and after a La Niña event. Suitable low temperatures and high primary productivity may promote the growth and development of plankton and more fish. Beibu Gulf salinity increases from northeast to southwest, and it is highest in the open sea. Salinity affects fish metabolism. Fish egg development and growth of juvenile fish are also strongly affected by changes in salinity [56]. A significant increase in salinity in the northeastern gulf caused by a La Niña event (Figure 8) may have produced an environment that was favorable for survival and growth of small pelagic fish. Concurrently, abundance in the central and southern waters increased. The effect of DO may work in concert with salinity and temperature. We report that the abundance of *T. japonicus* and *D. maruadsi* was mainly concentrated at approximately 50–75 m depth. This greater depth may be associated with reduced environmental variation, providing a more suitable habitat for these species [57]. Changes in SST, MLD, SSS, Pre, and Depth caused by El Niño events significantly affect *T. japonicus* and *D. maruadsi*. During winter, northern gulf temperatures are cooler than those in the south (Figure 10). During El Niño events, the Beibu Gulf sea area more greatly impacts SST through zonal wind and latent heat flux, and warming is obvious [58]. The MLD normally changes seasonally, and is most affected by dynamic and thermal factors on the sea surface. In normal years, MLD changes with the seasons, and the reduced energy of a weakened winter monsoon cannot maintain a deep spring MLD (Figure 11). Therefore, the depth of the mixed layer becomes shallow, and the distribution characteristics are still gradually deep from north to south. Additionally, the depth of the MLD becomes significantly shallower in the south–central Gulf in late winter. During an El Niño event, sea water disturbance weakens and the MLD becomes less variable in winter. Fish community overall catch trends downward during an El Niño event, and highly mobile *T. japonicus* and *D. maruadsi* stocks migrate to more favorable central and southern waters.

ENSO affects the distribution and abundance of fish species by changing their adaptability to their habitat. The intensity of effect depends on the type of event [5]. Changes in fish communities affect fisheries, and thus fisheries management policies must also adapt, because major fluctuations in fishery resources caused by climate change complicate estimation of allowable catch. Increased monitoring and investigation of fishery resources is necessary, scientific and technological support should be enhanced, and strong support for scientific implementation of resource management should be provided. At the same time, production statistics and information monitoring should be improved to better react to the dynamic state of fisheries, so that fishing structure and layout can be more appropriately adjusted.

## 5. Conclusions

We report that fish species richness and community structure during summer in Beibu Gulf vary between years, and both are affected by changes in environmental variables caused by ENSO events. Abundances of small pelagic fish such as *T. japonicus* and *D. maruadsi* fluctuate greatly, particularly during La Niña events. Ongoing climate change may result in more extreme water temperatures in Beibu Gulf, which may affect the growth, survival, and distribution of fish. La Niña environmental conditions bring cooler water temperatures and increased primary productivity, positively affecting fish communities. The impacts of dramatic and rapid changes in environmental conditions on fisheries resources brought about by ENSO events should be considered in fisheries management policies. Future research should determine whether our findings apply to other fish species, overcome the limitations of some models, and use measured data to verify model results.

## Figures and Tables

**Figure 1 biology-12-00946-f001:**
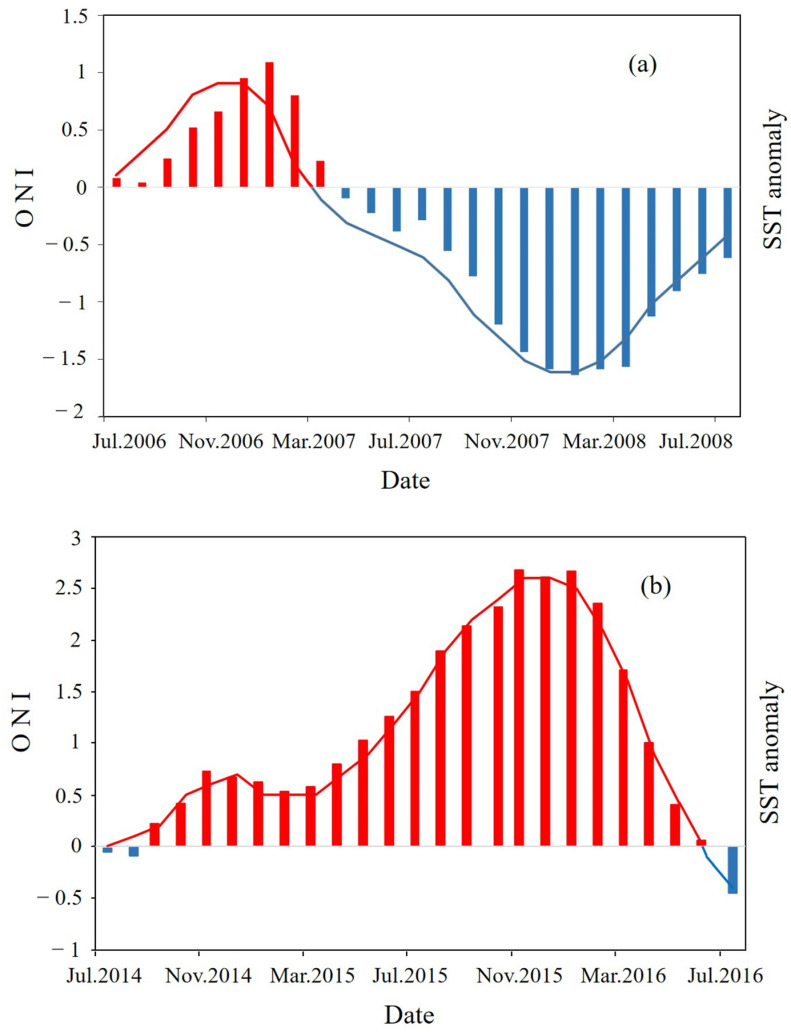
Monthly Oceanic Niño Index (ONI) and sea surface temperature (SST) anomalies in Beibu Gulf: (**a**) June 2007 to June 2008 (La Niña), (**b**) October 2014 to April 2016 (El Niño). Bars represent SST anomalies; lines represent the ONI index: blue (positive), red (negative).

**Figure 2 biology-12-00946-f002:**
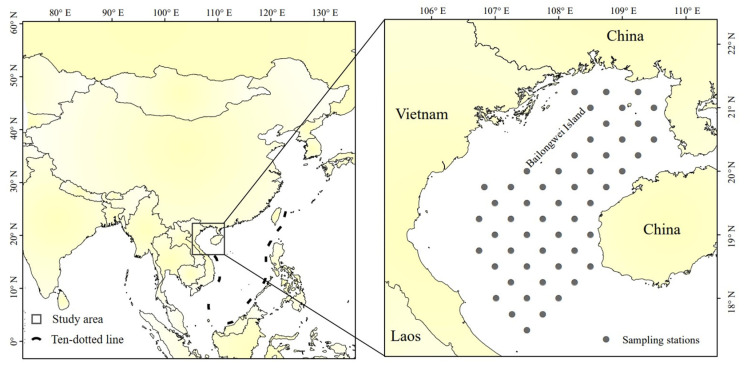
Beibu Gulf, and (inset) survey stations.

**Figure 3 biology-12-00946-f003:**
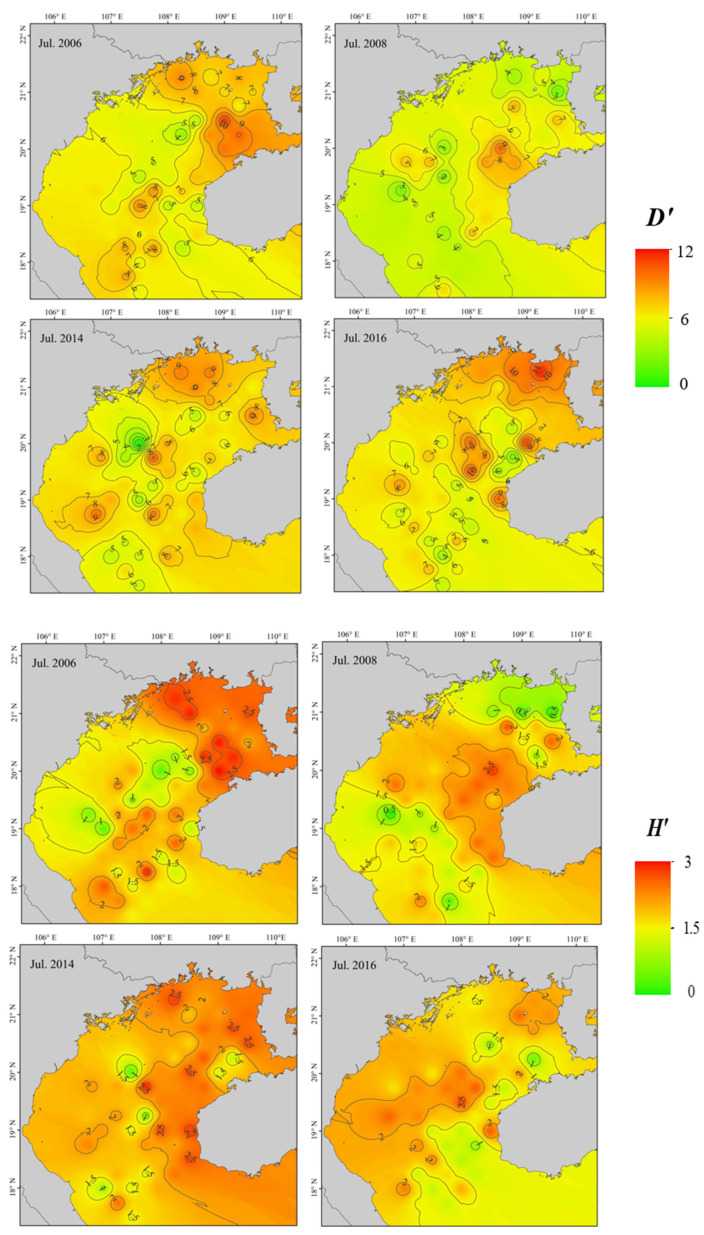
Pre- and post-ENSO diversity index (*D*′, *H*′, *J*′) distributions in Beibu Gulf.

**Figure 4 biology-12-00946-f004:**
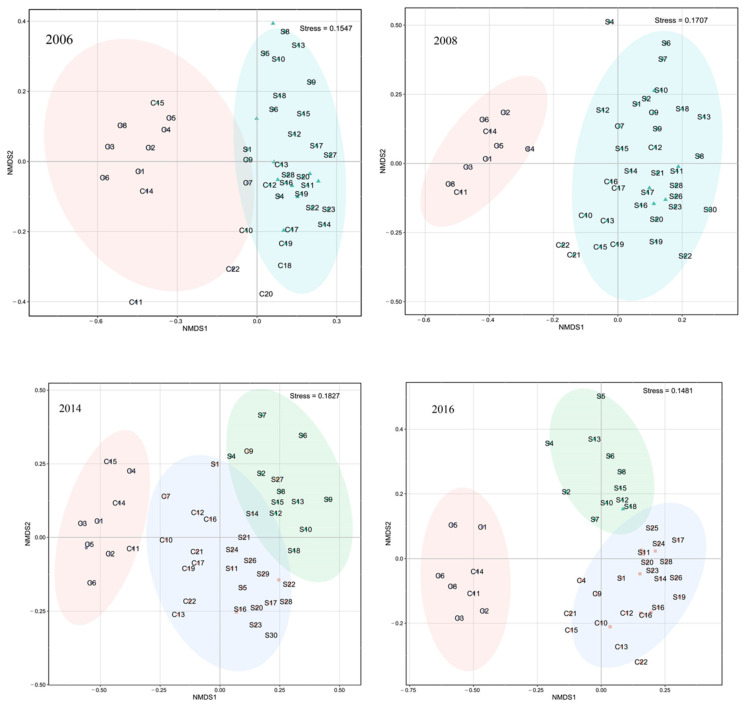
NMDS analysis of fish in Beibu Gulf pre- and post-ENSO (pre- and post-La Niña, summers of 2006 and 2008; pre- and post-El Niño, summers of 2014 and 2016).

**Figure 5 biology-12-00946-f005:**
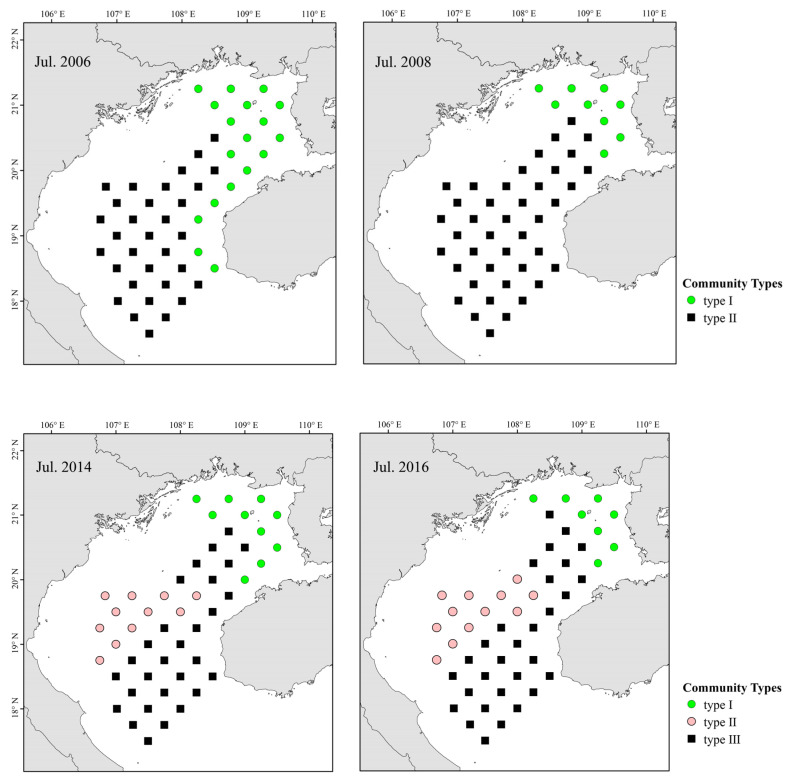
Spatial and temporal distributions of fish assemblages in Beibu Gulf.

**Figure 6 biology-12-00946-f006:**
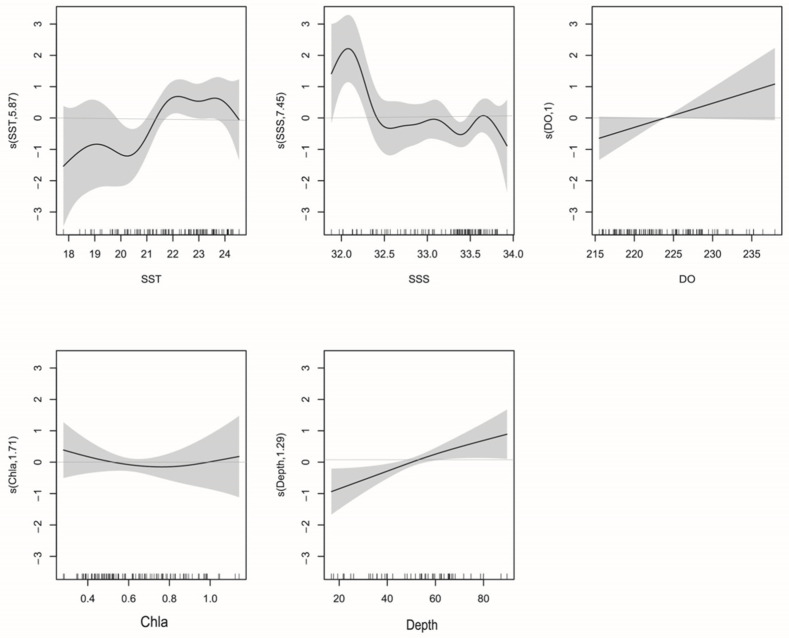
La Niña: GAM analysis of the impacts of environmental variables on *T. japonicus* and *D. maruadsi* resource abundance. The solid lines are the fitted curves and the shadows are the 95% confidence intervals.

**Figure 7 biology-12-00946-f007:**
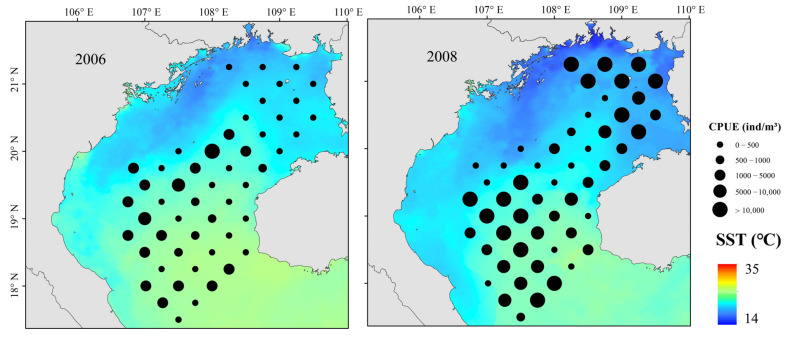
Spatial distributions of *T. japonicus* and *D. maruadsi* and SST during La Niña events, Beibu Gulf.

**Figure 8 biology-12-00946-f008:**
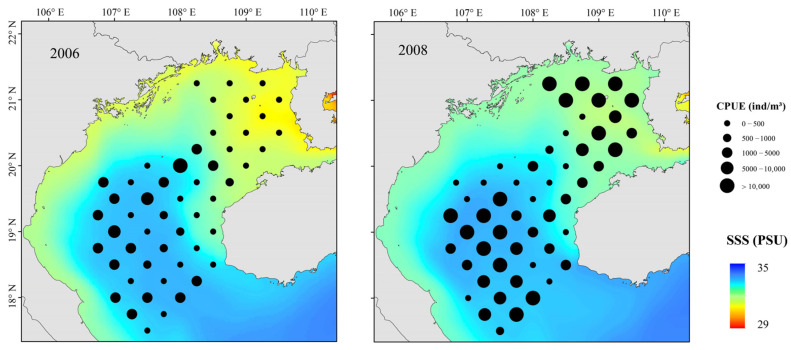
Spatial distributions of *T. japonicus* and *D. maruadsi* and SSS in La Niña event, Beibu Gulf.

**Figure 9 biology-12-00946-f009:**
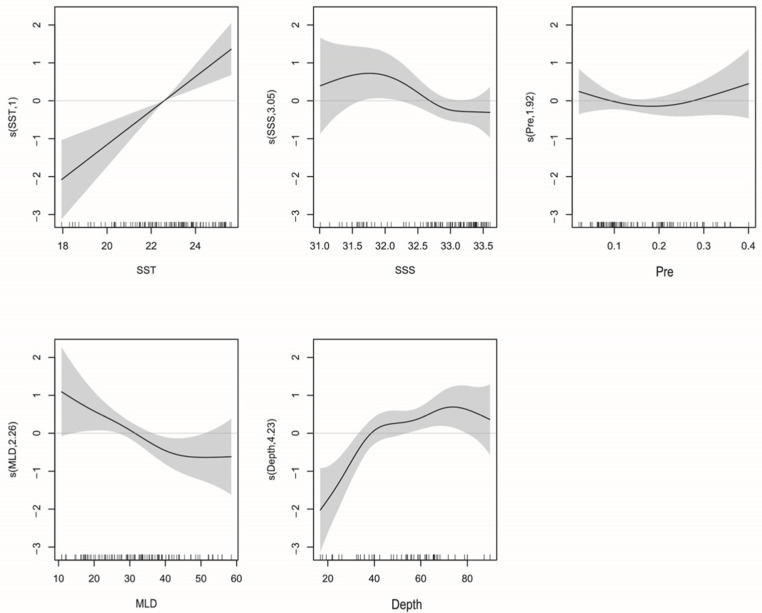
El Niño: GAM analysis of the impacts of environmental variables on resource abundance. The solid lines are the fitted curves and the shadows are the 95% confidence intervals.

**Figure 10 biology-12-00946-f010:**
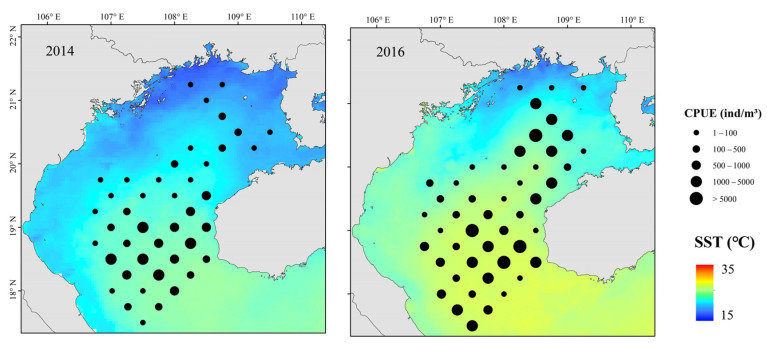
Spatial distributions of *T. japonicus* and *D. maruadsi* and SST during El Niño events, Beibu Gulf.

**Figure 11 biology-12-00946-f011:**
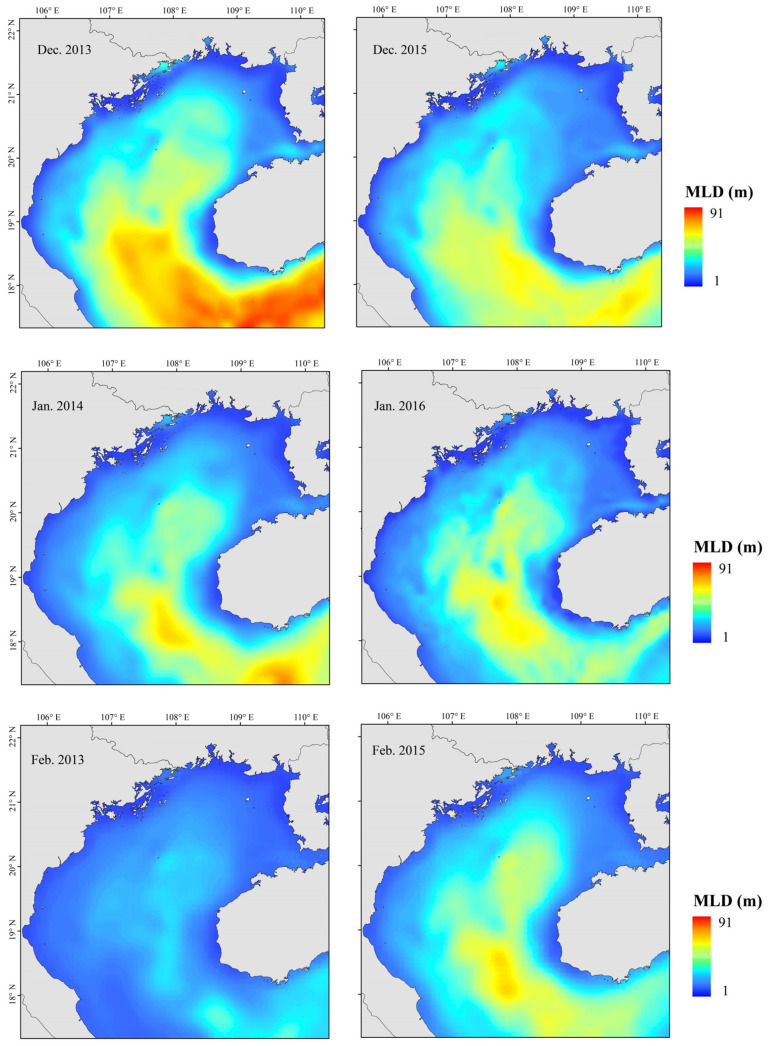
Mixed layer depth (MLD) in “normal” and El Niño years, Beibu Gulf. (Left: “normal” years; Right: El Niño years).

**Table 1 biology-12-00946-t001:** Sources of marine environmental data.

Variables (Unit)	Data Sources	Spatial Resolution	Download Website
SSS (PSU)	CMEMS	1/12° × 1/12°	https://Marine.Copernicus.eu (accessed on 10 July 2022)
DO (mg·m^−3^)	CMEMS	1/12° × 1/12°	https://Marine.Copernicus.eu (accessed on 10 July 2022)
MLD (m)	CMEMS	1/12° × 1/12°	https://Marine.Copernicus.eu (accessed on 10 July 2022)
Chl-a (mg·m^−3^)	CMEMS	4 km	https://Marine.Copernicus.eu (accessed on 10 July 2022)
SST (°C)	MODIS-Aqua	4 km	http://www.cpc.ncep.noaa.gov/ (accessed on 20 July 2022)
Pre (mm·h^−1^)	TRMM	5 km	https://disc.gsfc.nasa.gov/ (accessed on 20 July 2022)

**Table 2 biology-12-00946-t002:** Temporal changes in dominant fish species during summer in Beibu Gulf.

Month Year	Species	IRI	pIRI (%)
Summer 2006	*Acropoma japonicum*	4341.33	38.46
	*Trachurus japonicus*	3550.31	31.45
Summer 2008	*Trachurus japonicus*	7745.79	49.97
	*Decapterus maruadsi*	4457.01	28.75
	*Acropoma japonicum*	1607.03	10.37
Summer 2014	*Acropoma japonicum*	3918.11	33.08
	*Evynnis cardinalis*	2531.58	21.37
	*Trachurus japonicus*	2022.55	17.07
Summer 2016	*Trachurus japonicus*	5365.45	40.66
	*Acropoma japonicum*	2272.35	17.22
	*Psenopsis anomala*	1799.63	13.64
	*Evynnis cardinalis*	1587.05	12.03
	*Decapterus maruadsi*	1153.2	8.74

**Table 3 biology-12-00946-t003:** Analysis of deviation for the generalized additive model. AIC = Akaike information criterion; GCV = generalized cross-validation.

Event	Model Variables	*R* ^2^	Deviance Explained/%	AIC	GCV
La Niña	log(CPUE + 1) = s(Year) + s(SST)	0.353	39.7	317.50	1.22
	log(CPUE + 1) = s(Year) + s(SST) + s(Chl-a)	0.371	42.4	315.89	1.20
	log(CPUE + 1) = s(Year) + s(SST) + s(Chl-a) + s(Depth)	0.379	42.8	314.00	1.19
	log(CPUE + 1) = s(Year) + s(SST) + s(Chl-a) + s(Depth) + s(DO)	0.442	51.2	307.05	1.12
	log(CPUE + 1) = s(Year) + s(SST) + s(Chl-a) + s(Depth) + s(DO) + s(SSS)	0.553	63.2	288.58	0.96
El Niño	log(CPUE + 1) = s(Year) + s(SSS)	0.244	25.8	301.55	1.04
	log(CPUE + 1) = s(Year) + s(SSS) + s(Pre)	0.336	36.8	290.93	0.95
	log(CPUE + 1) = s(Year) + s(SSS) + s(Pre) + s(MLD)	0.387	43.1	284.89	0.90
	log(CPUE + 1) = s(Year) + s(SSS) + s(Pre) + s(MLD) + s(Depth)	0.522	57.9	263.15	0.73
	log(CPUE + 1) = s(Year) + s(SSS) + s(Pre) + s(MLD) + s(Depth) + s(SST)	0.571	62.7	253.05	0.69

## Data Availability

The datasets used and/or analyzed during the current study are available from the corresponding author upon reasonable request.

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
