# Peer review of "Impacts of Strong ENSO Events on Fish Communities in an Overexploited Ecosystem in the South China Sea"

_biology, 2023, doi:10.3390/biology12070946_

Round 1

Reviewer 1 Report

Title: Impacts of strong ENSO events on fish communities in an overexploited ecosystem, South China Sea

Li et al. studied the effects that climate change has on marine ecosystem and fish productivity in the Beibu Gulf. The study analyzed the Beibu Gulf fishery resource survey data before and after ENSO events, comparing the changes in fish biodiversity, dominant species and community structure spatial patterns between El Niño and La Niña events. The study further examines the relationship between the abundance of dominant species and environmental parameters, and hence, evaluates how environmental variables affect fish populations in the Beibu gulf.

The study being important for the proper management of fisheries in the Beibu gulf, I however, think some improvements in the MS can benefit clearer readability and understanding by readers. The language used throughout the MS would make the MS clearer if edited by a native speaker. Therefore, a couple of comments to be addressed by authors are listed below:

 My major concern lies in the data collection/ data length used:

MM. Survey data

Authors presented just survey data for July for every sampling year. I wonder if using just a month a year for all sampling years would be sufficient to carry out and present results that reflects the actual situation of fish production and effects of environment in the Beibu Gulf.

Moreover, I would have advised redirecting their writing towards “comparing different influences of environmental variables on fish production during summer period before and after nino events” in the study area if authors presented at least 2 months for summer; Just July does not actually represent summer.

Do authors have reasons for considering just a month (closure month) for each sampling year? Please justify because the argument in Lines 70-72 seems so light…

minor concers: 

Introduction

Lines 22-23: replace “change” with “changed” and “is” by “was”

Lines 25: … Japanese scad re…? I think it should be “were”

Materials and Methods

Lines 103-104: What is the importance of this sentence? What data authors are referring to here?

Line 137: Rephrase this sentence.

Line 170: any reference to back up this selection of VIF, since most studies suggest including values for VIF<5 and not 10?

Conclusion

I still don’t think that authors should use the conclusion of this paper to be a general conclusion for fish species in the Beibu Gulf, since only a month data was used for each sampling year. Rephrase or add more data (more months for each sampled year).

Figures

Figure 1: Wrong captions. Where do we see “a” or “b” on the figure. Also, explain what the blue and orange bars indicate, and what the blue and red lines means.

English language needs to be improved.

Author Response

Dear Reviewer,

We would like to thank you for your efforts in reviewing our manuscript titled " Impacts of strong ENSO events on fish communities in an overexploited ecosys-tem, South China Sea", and for providing many helpful comments and suggestions, which will all prove invaluable in the revision and improvement of our paper, as well as in guiding our research in the future. Your suggestion and guidance were accepted and to mark these changes using MS word's track changes feature. We appreciate your warm work earnestly and hope that the correction will meet with approval.

In the following, the points you mentioned will be discussed:

Authors presented just survey data for July for every sampling year. I wonder if using just a month a year for all sampling years would be sufficient to carry out and present results that reflects the actual situation of fish production and effects of environment in the Beibu Gulf. Moreover, I would have advised redirecting their writing towards “comparing different influences of environmental variables on fish production during summer period before and after nino events” in the study area if authors presented at least 2 months for summer; Just July does not actually represent summer. Do authors have reasons for considering just a month (closure month) for each sampling year? Please justify because the argument in Lines 70-72 seems so light…

Response: We apologise for not expressing clearly the reasons for the selection of the data. Firstly, the breeding period for small pelagic species in the Beibu Gulf, represented by Japanese jack mackerel (Trachurus japonicus) and Japanese scad (Decapterus maruadsi), is mainly in winter. We believe that ENSO events affect their replenishment processes primarily through altered environmental factors and that the use of resource distribution and density data for July provides a visual indication of the impact of their winter replenishment. Such impacts from ENSO events are lagged by topographic constraints, biological interactions, and species characteristics.[6-10] T. japonicus and D. maruadsi, being current year fish, habitat changes can affect fish replenishment and this lag will be reflected in the current year. Secondly, the reason we used the July survey data was to reduce the interference of fishing on the results. This is because July falls during the vadose season closure in the South China Sea region when fishing intensity is negligible. Finally, as July is not a month that represents summer, we have changed the text with summer in it to July, as you suggested.

Introduction

Lines 22-23: replace “change” with “changed” and “is” by “was”

Response: Thank you for your suggestion, we have modified this.

Lines 25: … Japanese scad re…? I think it should be “were”

Response: Thank you for your suggestion, and we apologize for our lax expression.

Materials and Methods

Lines 103-104: What is the importance of this sentence? What data authors are referring to here?

Response: We are very sorry that we did not express ourselves clearly. July is the closed season for fishing in the South China Sea area, so fishing efforts in the Beibu Gulf can be considered manageable and negligible at this time. Ultimately, the main consideration is the impact of climate change.

Line 137: Rephrase this sentence.

Response: Thank you for your suggestion. We apologize for the poor presentation here and have now revised the language here in the manuscript. Species were divided by dominance:IRI ≥ 1000 (dominant), 500 ≤ IRI < 1000 (important), 100 ≤ IRI < 500 (common), 10 ≤ IRI < 100 (uncommon), and IRI < 10 (rare).

Line 170: any reference to back up this selection of VIF, since most studies suggest including values for VIF<5 and not 10?

Response: We are very sorry to have caused you a problem. The explanatory variables were tested for multicollinearity by VIF to exclude highly correlated explanatory variables. There is no unified conclusion on the threshold value of VIF, and the scientific studies that have been conducted are largely based on previous research and their own experience, with some thresholds being VIF < 5, others VIF < 10, and still others VIF < 7. Therefore, there is a scientific basis for our use of VIF < 10 as a threshold to assess covariance.

Conclusion

I still don’t think that authors should use the conclusion of this paper to be a general conclusion for fish species in the Beibu Gulf, since only a month data was used for each sampling year. Rephrase or add more data (more months for each sampled year).

Response: We apologise for not expressing clearly the reasons for the selection of the data. As July is not a month that represents summer, we have changed the text with summer in it to July, as you suggested. In our resource survey, we found that there were mostly significant fluctuations in fishery resources in the Beibu Gulf during the various July cruises before and after ENSO events over the past two decades. To explore this, we selected representative July survey data before and after strong La Niña events (July 2006 and 2008) and before and after strong El Niño events (July 2014 and 2016), respectively. Developing good management policies to address the impacts of extreme weather events on fish community structure in the Beibu Gulf.

Figures

Figure 1: Wrong captions. Where do we see “a” or “b” on the figure. Also, explain what the blue and orange bars indicate, and what the blue and red lines means.

Response: We are very sorry that we did not express ourselves clearly. We have modified the color band selection and legend labeling of Figure 1. The bar chart characterises the SST anomaly, with the blue and orange bars indicating negative and positive sea surface temperature anomalies, respectively. The folded line characterises the ONI index, with the blue and red lines indicating negative and positive ONI indices, respectively. And now we have reinserted the manuscript

Comments on the Quality of English Language: English language needs to be improved.

Response: We invited native an English speaking expert to polish the paper, please check the proof of polishing of the paper.

Reviewer 2 Report

Comments: 

This study examined the changes of the fish community in the Beibu Gulf during strong El Nino and La Nina events, providing a basis for improving scientific management measures. They suggest that strong La Nina and El Nino events affect the composition, abundance, and distribution of fish communities in Beibu Gulf. This paper deals with relevant and interesting topic and is potentially a valuable contribution to studies about effects of climate changes on marine fish communities. However, there are numerous problems, some of them major, that need to be addressed. The following is my primary concerns.

1. The authors need to better describe the specific objectives in this study in the introduction. Otherwise, we cannot know what this study was conducted for.

2. I recommend that the authors describe the hypotheses and predictions we can follow on this study in the introduction of this paper. 

3. The authors need to better describe about the statistical analyses used for this study. There are numerous statistical models (e.g., GLM, GAM, etc.) in the world. I am wondering why the authors used GAM to determine the effects of the climate changes.

4. The discussion seems to be out of focus possibly because the specific objectives in the introduction were not well explained.

5. In the conclusion, the authors suggest that “ENSO events do not seem to play a major role in environmental change on their long–time scales.” However, how can you conclude this? You need long-term fisheries data (e.g., >30 years) to conclude this.

6. The conservation/management suggestions are limited in this study. I recommend that the authors write the more specific conservation/management strategies based on your results.

7. There are numerous grammatical mistakes in this manuscript. Please check English grammar throughout the manuscript.

Author Response

Dear Reviewer,

We would like to thank you for your efforts in reviewing our manuscript titled " Impacts of strong ENSO events on fish communities in an overexploited ecosys-tem, South China Sea", and for providing many helpful comments and suggestions, which will all prove invaluable in the revision and improvement of our paper, as well as in guiding our research in the future. Your suggestion and guidance were accepted and to mark these changes using MS word's track changes feature. We appreciate your warm work earnestly and hope that the correction will meet with approval.

In the following, the points you mentioned will be discussed:

  1. The authors need to better describe the specific objectives in this study in the introduction. Otherwise, we cannot know what this study was conducted for.

Response: Thanks to your suggestions, we have revised the introduction section to better describe the specific objectives of this study. In the context of global warming, extreme climate change events bring great uncertainty to fisheries management. By analysing survey data to reveal the mechanisms leading to fluctuations in fishery resources in the Beibu Gulf, the impact of extreme climate events on a typical overfished bay ecosystem in the South China Sea needs to be explored to provide a scientific basis for further improving the management and conservation regime of fishery resources in the Beibu Gulf.

  1. I recommend that the authors describe the hypotheses and predictions we can follow on this study in the introduction of this paper.

Response: Thank you for your suggestion, we have revised the introductory section to add our assumptions. ENSO events have had a large impact on sea level elevation, sea surface temperature, primary productivity and ocean circulation in the South China Sea.[27-29] As a result, changes in habitat will likely affect the growth, mortality, fecundity, spawning distribution and movement of fish in the Beibu Gulf. In particular, small pelagic fishes, which occupy an important ecological niche in the Beibu Gulf, are highly sensitive to climate change. Therefore, we used our study to determine the composition, structure, diversity, abundance, and community changes of fish communities in the northern Gulf before and after the ENSO event.

  1. The authors need to better describe about the statistical analyses used for this study. There are numerous statistical models (e.g., GLM, GAM, etc.) in the world. I am wondering why the authors used GAM to determine the effects of the climate changes.

Response: We are very sorry that we did not express it clearly. The relationship between the spatial and temporal distribution of fishery resources and environmental variables is very complex, non-linear, and non-additive. [36] Generalized additive models can better handle the relationship between response variables and multiple explanatory variables, have a high degree of flexibility in model construction, and are therefore widely used in the qualitative analysis of the relationship between fisheries resources and the marine environment. [37,38]

  1. The discussion seems to be out of focus possibly because the specific objectives in the introduction were not well explained.

Response: We are very sorry that we did not express it clearly. Because of the increasing number of extreme climate change events in the context of global warming, there is a great deal of uncertainty in fisheries management. By analyzing survey data to reveal the mechanisms of fishery resource fluctuations in the Beibu Gulf, the impact of extreme climatic events on typical overfished bay ecosystems in the South China Sea is explored. By using survey data from the fishing moratorium phase, the interference of fishing in the results is reduced. Our discussion of the article begins with the state of change in fish community structure in the Beibu Gulf caused by the ENSO event to reveal the mechanisms of fluctuations in fishery resources in the Beibu Gulf. We also found that small pelagic fish species, represented by T. japonicus and D. maruadsi, fluctuated most significantly, and their population dynamics were highly sensitive to climate change, leading to significant changes in their resources. We therefore also analyzed the spatial and temporal distribution characteristics and influencing factors of typical pelagic fish resources in the context of habitat change, and explored the impact of extreme climatic events on the Beibu Gulf ecosystem.

  1. In the conclusion, the authors suggest that “ENSO events do not seem to play a major role in environmental change on their long–time scales.” However, how can you conclude this? You need long-term fisheries data (e.g., >30 years) to conclude this.

Response: We apologize for the confusion our manuscript has caused you. Thank you for your suggestion, we have revised this part.

.

  1. The conservation/management suggestions are limited in this study. I recommend that the authors write the more specific conservation/management strategies based on your results.

Response: I'm sorry we didn't express ourselves clearly. Because changes in fish communities affect fisheries, fisheries management policies must be able to adapt to them. Drastic fluctuations in fishery resources caused by climate change may cause great difficulties in estimating the allowable catch. We should strengthen the monitoring and investigation of fishery resources, enhance scientific and technological support, and provide strong support for the scientific implementation of resource management. At the same time, we should improve production statistics and information monitoring, to grasp the dynamics of fishery production in a timely and accurate manner and provide a basis for reasonable adjustment of fishing structure and fishing layout. This part has been added to the manuscript, and we thank you for your suggestions.

  1. There are numerous grammatical mistakes in this manuscript. Please check English grammar throughout the manuscript.

Response: Thank you for your comments, we invited native an English speaking expert to polish the paper, please check the proof of polishing of the paper.

Round 2

Reviewer 1 Report

The authors have expressed clearly my worries. no further comments.

Author Response

Thank you again for your valuable suggestions to improve the quality of our manuscript. 

Reviewer 2 Report

I think that the authors appropriately respond to my comments. I recommend that the authors check the manuscript again to make sure if there are minor mistakes in the manuscript. Thank you for giving me the opportunity for reviewing this manuscript.

Author Response

Thank you again for your valuable suggestions to improve the quality of our manuscript. We have check the whole manuscript to make sure there are no mistakes.